# Hyperbolic Graph Neural Networks at Scale: A Meta Learning Approach

**Nurendra Choudhary**
Virginia Tech
Arlington, VA, USA
nurendra@vt.edu

**Nikhil Rao**
Microsoft
Sunnyvale, CA, USA
nikhilrao@microsoft.com

**Chandan K. Reddy**
Virginia Tech
Arlington, VA, USA
reddy@cs.vt.edu

## Abstract

The progress in hyperbolic neural networks (HNNs) research is hindered by their absence of inductive bias mechanisms, which are essential for generalizing to new tasks and facilitating scalable learning over large datasets. In this paper, we aim to alleviate these issues by learning generalizable inductive biases from the nodes' local subgraph and transfer them for faster learning over new subgraphs with a disjoint set of nodes, edges, and labels in a few-shot setting. We introduce a novel method, Hyperbolic GRAph Meta Learner (H-GRAM), that, for the tasks of node classification and link prediction, learns transferable information from a set of support local subgraphs in the form of hyperbolic meta gradients and label hyperbolic protonets to enable faster learning over a query set of new tasks dealing with disjoint subgraphs. Furthermore, we show that an extension of our meta-learning framework also mitigates the scalability challenges seen in HNNs faced by existing approaches. Our comparative analysis shows that H-GRAM effectively learns and transfers information in multiple challenging few-shot settings compared to other state-of-the-art baselines. Additionally, we demonstrate that, unlike standard HNNs, our approach is able to scale over large graph datasets and improve performance over its Euclidean counterparts.

## 1 Introduction

Graphs are extensively used in various applications, including image processing, natural language processing, chemistry, and bioinformatics. With modern graph datasets ranging from hundreds of thousands to billions of nodes, research in the graph domain has shifted towards larger and more complex graphs, e.g., the nodes and edges in the traditional Cora dataset [29] are in the order of $10^3$, whereas, the more recent OGBN datasets [17] are in the order of $10^6$. However, despite their potential, Hyperbolic Neural Networks (HNNs), which capture the hierarchy of graphs in hyperbolic space, have struggled to scale up with the increasing size and complexity of modern datasets.

HNNs have outperformed their Euclidean counterparts by taking advantage of the inherent hierarchy present in many modern datasets [4, 10, 13]. Unlike a standard Graph Neural Network (GNN), a HNN learns node representations based on an "anchor" or a "root" node for the entire graph, and the operations needed to learn these embeddings are a function of this root node. Specifically, HNN formulations [13] depend on the global origin (root node) for several transformation operations (such as Möbius addition, Möbius multiplication, and others), and hence focusing on subgraph structures to learn representations becomes meaningless when their relationship to the root node is not considered. Thus, state-of-the-art HNNs such as HGCN [4], HAT [15], and HypE [8] require access to the entire dataset to learn representations, and hence only scale to experimental datasets (with $\approx 10^3$ nodes). Despite this major drawback, HNNs have shown impressive performance on several research domains including recommendation systems [32], e-commerce [9], natural language processing [11], and knowledge graphs [5, 8]. It is thus imperative that one develops methods to scale HNNs to larger datasets, so as to realize their full potential. To this end, we introduce a novel method, **Hyperbolic GRAph Meta Learner (H-GRAM)**, that utilizes meta-learning to learn infor-

37th Conference on Neural Information Processing Systems (NeurIPS 2023).

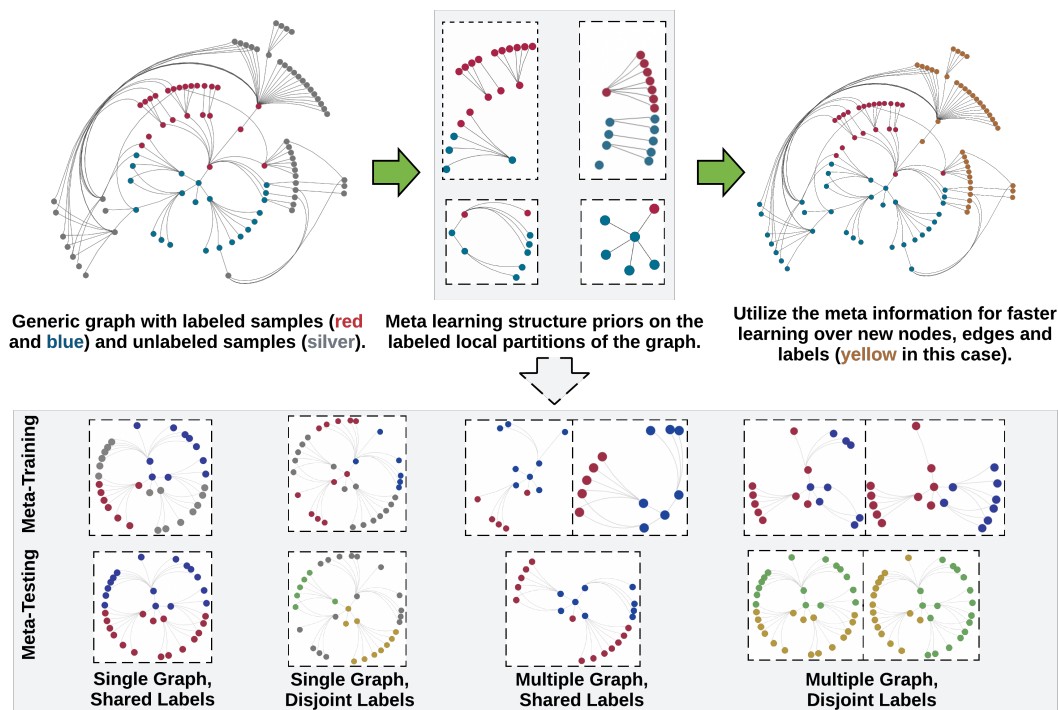

**Generic graph with labeled samples (red and blue) and unlabeled samples (silver).**

**Meta learning structure priors on the labeled local partitions of the graph.**

**Utilize the meta information for faster learning over new nodes, edges and labels (yellow in this case).**

Meta-Training

Meta-Testing

**Single Graph, Shared Labels**  **Single Graph, Disjoint Labels**  **Multiple Graph, Shared Labels**  **Multiple Graph, Disjoint Labels**

Figure 1: Meta-learning on hyperbolic neural networks. The procedure consists of two phases - (i) meta-training to update the parameters of the HNNs and learn inductive biases (meta gradients and label protonets), and (ii) meta-testing that initializes the HNNs with the inductive biases for faster learning over new graphs with a disjoint set of nodes, edges, or labels.

mation from local subgraphs for HNNs and transfer it for faster learning on a disjoint set of nodes, edges, and labels contained in the larger graph. As a consequence of meta-learning, H-GRAM also achieves several desirable benefits that extend HNNs' applicability including the ability to transfer information on new graphs (inductive learning), elimination of over-smoothing, and few-shot learning. We experimentally show that H-GRAM can scale to graphs of size $\mathcal{O}(10^6)$, which is $\mathcal{O}(10^3)$ times larger than previous state-of-the-art HNNs. To the best of our knowledge, there are no other methods that can scale HNNs to datasets of this size.

Recent research has shown that both node-level and edge-level tasks only depend on the local neighborhood for evidence of prediction [18, 19, 42]. Inspired by the insights from such research, our model handles large graph datasets using their node-centric subgraph partitions, where each subgraph consists of a root node and the $k$-hop neighborhood around it. In H-GRAM, the HNN formulations establish the root node as the local origin to encode the subgraph. We theoretically show that, due to the locality of tangent space transformations in HNNs (more details in Section 3), the evidence for a node's prediction can predominantly be found in the immediate neighborhood. Thus, the subgraph encodings do not lose a significant amount of feature information despite not having access to a "global" root node. However, due to the node-centric graph partitioning, the subgraphs are non-exhaustive, i.e., they do not contain all the nodes, edges, and labels, as previously assumed by HNNs. Thus, to overcome the issue of non-exhaustive subgraphs, we formulate four meta-learning problems (illustrated in Figure 1) that learn inductive biases on a support set and transfer it for faster learning on a query set with disjoint nodes, edges, or labels. Our model learns inductive biases in the meta-training phase, which contains two steps - HNN update and meta update. HNN updates are regular stochastic gradient descent steps based on the loss function for each support task. The updated HNN parameters are used to calculate the loss on query tasks and the gradients are accumulated into a meta-gradient for the meta update. In the meta-testing phase, the models are evaluated on query tasks with parameters post the meta updates, as this snapshot of the model is the most adaptable for faster learning in a few-shot setting. Our main contributions are as follows:

1. We theoretically prove that HNNs rely on the nodes' local neighborhood for evidence in prediction, as well as, formulate HNNs to encode node-centric local subgraphs with root nodes as the local origin using the locality of tangent space transformations.
2. We develop Hyperbolic GRAph Meta Learner (H-GRAM), a novel method that learns meta information (as meta gradients and label protonets) from local subgraphs and generalize it to new graphs with a disjoint set of nodes, edges, and labels. Our experiments show that H-GRAM can be used to generalize information from subgraph partitions, thus, enabling scalability.
3. Our analysis on a diverse set of datasets demonstrates that our meta-learning setup also solves several challenges in HNNs including inductive learning, elimination of over-smoothing, and few-shot learning in several challenging scenarios.

## 2    Related Work

This section reviews the relevant work in the areas of hyperbolic neural networks and meta learning.

**Hyperbolic Neural Networks:** Due to their ability to efficiently encode tree-like structures, hyperbolic space has been a significant development in the modeling of hierarchical datasets [43, 44]. Among its different isometric interpretations, the Poincaré ball model is the most popular one and has been applied in several HNN formulations of Euclidean networks including the recurrent (HGRU [13]), convolution (HGCN [4]), and attention layers (HAT [15]). As a result of their performance gains on hierarchical graphs, the formulations have also been extended to applications in knowledge graphs for efficiently encoding the hierarchical relations in different tasks such as representation learning (MuRP [1], AttH [5]) and logical reasoning (HypE [8]). However, the above approaches have been designed for experimental datasets with a relatively small number of nodes (in the order of $10^3$), and do not scale to real-world datasets. Hence, we have designed H-GRAM as a meta-learning algorithm to translate the performance gains of HNNs to large datasets in a scalable manner.

**Graph Meta-learning:** Few-shot meta-learning transfers knowledge from prior tasks for faster learning over new tasks with few labels. Due to their wide applicability, they have been adopted in several domains including computer vision [14, 34], natural language processing [22] and, more recently, graphs [18, 39]. One early approach in graph meta-learning is Gated Propagation Networks [23] which learns to propagate information between label prototypes to improve the information available while learning new related labels. Subsequent developments such as MetaR [7], Meta-NA [41] and G-Matching [39] relied on metric-based meta learning algorithms for relational graphs, network alignment and generic graphs, respectively. These approaches show impressive performance on few-shot learning, but are only defined for single graphs. G-Meta [18] extends the metric-based techniques to handle multiple graphs with disjoint labels. However, the method processes information from local subgraphs in a Euclidean GNN, and thus, is not as capable as hyperbolic networks in encoding tree-like structures. Thus, we model H-GRAM to encode hierarchical information from local subgraphs and transfer it to new subgraphs with disjoint nodes, edges, and labels.

## 3    The Proposed H-GRAM Model

In this section, we define the problem setup for different meta-learning scenarios and describe our proposed model, H-GRAM, illustrated in Figure 2. For better clarity, we explain the problem setup for node classification and use HGCN as the exemplar HNN model. However, the provided setup can be extended to link prediction or to other HNN models, which we have evaluated in our experiments. The preliminaries of hyperbolic operations and meta-learning are provided in Appendix A.

### 3.1    Problem Setup

Our problem consists of a group of tasks $\mathcal{T}_i \in \mathcal{T}$ which are divided into a support set $\mathcal{T}^s$ and query set $\mathcal{T}^q$, where $\mathcal{T}^s \cap \mathcal{T}^q = \phi$. Furthermore, each task $\mathcal{T}_i$ is a batch of node-centric subgraphs $S_u$ with a corresponding label $Y_u$ (class of root node in node classification or root link in link prediction). The subgraphs $S_u$ could be the partitions derived from a single graph or multiple graphs, both denoted by $\mathcal{G}^\cup = \mathcal{G}^s \cup \mathcal{G}^q$. We also define $Y_s = \{Y_u \in \mathcal{T}^s\}$ and $Y_q = \{Y_u \in \mathcal{T}^q\}$ as the set of labels in the support and query set respectively. The primary goal of meta-learning is to learn a predictor using the support set, $P_{\theta_*}(Y_s|\mathcal{T}^s)$ such that the model can quickly learn a predictor $P_\theta(Y_s|\mathcal{T}^s)$ on the query set. Following literature in this area [18], the problem categories are defined as follows:

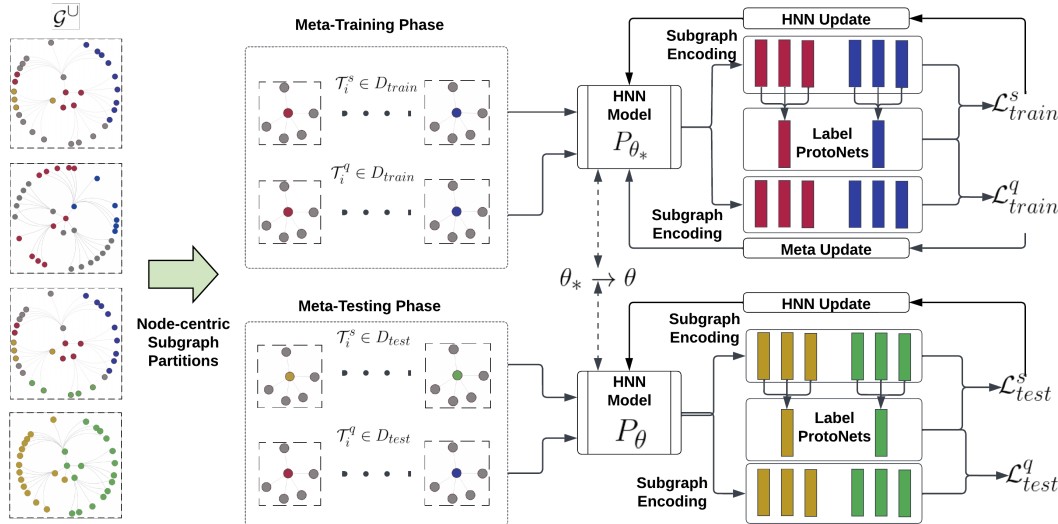

Figure 2: An overview of the proposed H-GRAM meta-learning framework. Here, the input graphs $\mathcal{G}^{\cup}$ are first partitioned into node-centric subgraph partitions. We theoretically show that encoding these subgraph neighborhoods is equivalent to encoding the entire graph in the context of node classification and link prediction tasks. H-GRAM then uses an HGCN encoder to produce subgraph encodings, which are further utilized to get label prototypes. Using the HGCN gradient updates and label prototypes, the HNN model's parameters $P_{\theta_*}$ is updated through a series of weight updates and meta updates for $\eta$ meta-training steps. The parameters are then transferred to the meta-testing stage $P_{\theta_* \to \theta}$ and further trained on $\mathcal{D}^s_{test}$ and evaluated on $\mathcal{D}^q_{test}$.

1. **Single graph, shared labels $\langle SG, SL \rangle$:** The objective is to learn the meta-learning model $P_{\theta_* \to \theta}$, where $Y_s = Y_q$ and $|\mathcal{G}^s| = |\mathcal{G}^q| = 1$. It should be noted that the tasks are based on subgraphs, so $|\mathcal{G}^s| = |\mathcal{G}^q| = 1 \implies |\mathcal{T}^s| = |T^q| = 1$. Also, this problem setup is identical to the standard node classification task considering $\mathcal{T}^q_i \in D_{test}$ to be the test set.
2. **Single graph, disjoint labels $\langle SG, DL \rangle$:** This problem operates on the same graph in the support and query set, however unlike the previous one, the label sets are disjoint. The goal is to learn the model $P_{\theta_* \to \theta}$, where $|\mathcal{G}^s| = |\mathcal{G}^q| = 1$ and $Y_s \cap Y_q = \phi$.
3. **Multiple graphs, shared labels $\langle MG, SL \rangle$:** This problem setup varies in terms of the dataset it handles, i.e., the dataset can contain multiple graphs instead of a single one. However, our method focuses on tasks which contain node-centric subgraphs, and hence, the model's aim is the same as problem 1. The aim is to learn the predictor model $P_{\theta_* \to \theta}$, where $Y_s = Y_q$ and $|\mathcal{G}^s|, |\mathcal{G}^q| > 1$.
4. **Multiple graphs, disjoint labels $\langle MG, DL \rangle$:** In this problem, the setup is similar to the previous one, but only with disjoint labels instead of shared ones, i.e., learn a predictor model $P_{\theta_* \to \theta}$, where $Y_s \cap Y_q = \phi$ and $|\mathcal{G}^s|, |\mathcal{G}^q| > 1$.

From the problem setups, we observe that, while they handle different dataset variants, the base HNN model operates on the $(S_u, Y_u)$ pair. So, we utilize a hyperbolic subgraph encoder and prototypical labels to encode $S_u$ and get a continuous version of $Y_u$ for our meta-learning algorithm, respectively.

### 3.2 Hyperbolic Subgraph Encoder

In previous methods [16, 18, 38], authors have shown that nodes' local neighborhood provides some informative signals for the prediction task. While the theory is not trivially extensible, we use the local tangent space of Poincaré ball model to prove that the local neighborhood policy holds better for HNN models. The reason being that, while message propagation is linear in Euclidean GNNs [42], it is exponential in HNNs. Hence, a node's influence, as given by Definition 1, outside its neighborhood decreases exponentially.

**Definition 1.** *The influence of a hyperbolic node vector $x^{\mathbb{H}}_v$ on node $x^{\mathbb{H}}_u$ is defined by the influence score $\mathcal{I}_{uv} = exp^c_0 \left( \left\| \frac{\partial \log^x_0(x^{\mathbb{H}}_u)}{\partial \log^x_0(x^{\mathbb{H}}_v)} \right\| \right).$*

**Definition 2.** *The influence of a graph $\mathcal{G}$ with set of vertices $\mathcal{V}$ on a node $u \in \mathcal{V}$ is defined as $\mathcal{I}_{\mathcal{G}}(u) = exp_0^c \left( \left( \sum_{v \in \mathcal{V}} log_0^c (\mathcal{I}_{uv}) \right) \right)$.*

**Theorem 1.** *For a set of paths $P_{uv}$ between nodes $u$ and $v$, let us define $D_{g\mu}^{p_i}$ as the geometric mean of nodes' degree in a path $p_i \in P_{uv}$, $p_{uv}$ as the shortest path, and $\mathcal{I}_{uv}$ as the influence of node $v$ on $u$. Also, let us say $D_{g\mu}^{min} = min \left\{ D_{g\mu}^{p_i} \forall p_i \in P_{uv} \right\}$, then the relation $\mathcal{I}_{uv} \leq exp_u^c \left( K / \left( D_{g\mu}^{min} \right)^{\|p_{uv}\|} \right)$ (where $K$ is a constant) holds for message propagation in HGCN models.*

Theorem 1 shows that the influence of a node decreases exponent-of-exponentially ($exp_u^c = \mathcal{O}(e^n)$) with increasing distance $\|p_{uv}\|$. Thus, we conclude that encoding the local neighborhood of a node is sufficient to encode its features for label prediction.

**Definition 3.** *The information loss between encoding an entire graph $\mathcal{G}$ and a subgraph $S_u$ with root node $u$ is defined as $\delta_{\mathbb{H}}(\mathcal{G}, S_u) = exp_0^c \left( log_0^c (\mathcal{I}_{\mathcal{G}}(u)) - log_0^c (\mathcal{I}_{S_u}(u)) \right)$.*

**Theorem 2.** *For a subgraph $S_u$ of graph $\mathcal{G}$ centered at node $u$, let us define a node $v \in \mathcal{G}$ with maximum influence on $u$, i.e., $v = \arg\max_t(\{\mathcal{I}_u t, t \in \mathcal{N}(u) \setminus u\})$. For a set of paths $P_{uv}$ between nodes $u$ and $v$, let us define $D_{g\mu}^{p_i}$ as the geometric mean of degree of nodes in a path $p_i \in P_{uv}$, $\|p_{uv}\|$ is the shortest path length, and $D_{g\mu}^{min} = min \left\{ D_{g\mu}^{p_i} \forall p_i \in P_{uv} \right\}$. Then, the information loss is bounded by $\delta_{\mathbb{H}}(\mathcal{G}, S_u) \leq exp_u^c \left( K / \left( D_{g\mu}^{min} \right)^{\|p_{uv}\|+1} \right)$ (where $K$ is a constant).*

Theorem 2 shows that encoding the local subgraph is a $e^{\|p_{uv}\|}$ order approximation of encoding the entire graph, and thus, with high enough $\|p_{uv}\|$, the encodings are equivalent. Note that $\|p_{uv}\|$ is equivalent to the subgraph's neighborhood size ($k$ in $k$-hop). This shows that encoding a node's local neighborhood is sufficient to encode its features. Theorem proofs are provided in Appendix B. The above theorems provide us with the theoretical justification for encoding local subgraphs into our meta-learning framework. Hence, we partition the input $\mathcal{G}^{\cup}$ into subgraphs $S_u = V \times E$ centered at root node $u$, with $|V|$ nodes, $|E|$ edges, a neighborhood size $k$, and the corresponding label $Y_u$. The subgraphs are processed through an $k$-layer HGCN network and the Einstein midpoint [33] of all the node encodings are taken as the overall encoding of the subgraph $S_u$. For a given subgraph $S_u = (A, X)$, where $A \in \mathbb{H}^{\|V\| \times \|V\|}$ is the local adjacency matrix and $X \in \mathbb{H}^{\|V\| \times m}$ are the node feature vectors of $m$ dimension, the encoding procedure given can be formalized as:

$$h_u = HGCN_{\theta^*}(A, X), \text{ where } h_u \in \mathbb{H}^{\|V\| \times d} \tag{1}$$

$$e_u = \frac{\sum_{i=1}^{\|V\|} \gamma_{iu} h_{iu}}{\sum_{i=1}^{\|V\|} \gamma_{iu}}, \text{ where } \gamma_{iu} = \frac{1}{\sqrt{1 - \|h_{iu}\|^2}} \tag{2}$$

where $HGCN(A, X) \in \mathbb{H}^{\|V\| \times d}$ is the output of k-layer HGCN with $d$ output units and $e_u \in \mathbb{H}^d$ is the final subgraph encoding of $S_u$. $\gamma_{iu}$ is the Lorentz factor of the hyperbolic vector $h_{iu}$ that indicates its weightage towards the Einstein midpoint.

### 3.3 Label Prototypes

Label information is generally categorical in node classification tasks. However, this does not allow us to pass inductive biases from the support set to the query set. Hence, to circumvent this issue, we use prototypical networks [31] as our label encoding. Our approach constructs continuous label prototypes by using the mean of meta-training nodes' features that belong to the label. These prototypes are then employed to classify meta-testing samples based on their similarity to the corresponding meta-training label prototypes. This enables our model to handle new, non-exhaustive labels in an inductive manner, without the need for additional training data. The primary idea is to form continuous label prototypes using the mean of nodes that belong to the label. To this end, the continuous label prototype of a label $y_k$ is defined as $c_k = \frac{\sum_{Y_u=y_k} \gamma_i e_i}{\sum_{Y_u=y_k} \gamma_i}$, where $\gamma_i = \frac{1}{\sqrt{1-\|e_i\|^2}}$, and $e_i \in \mathbb{H}^d$ is encoding of subgraphs $S_u$ with labels $Y_u = y_k$. For each $S_u$ with class $y_k$, we compute the class distribution vector as $p_k = \frac{e^{(-d_{\mathbb{H}}^c(e_u, c_k))}}{\sum_k e^{(-d_{\mathbb{H}}^c(e_u, c_k))}}$, where $d_{\mathbb{H}}^c(e_u, c_k) = \frac{2}{\sqrt{c}} \tanh^{-1} \left( \sqrt{c} \| -e_u \oplus c_k \| \right)$ and the loss for HNN updates $\mathcal{L}(p, y) = \sum_j y_i \log p_j$, where $y_i$ is the one-hot encoding of the ground truth. The class distribution vector $p_k$ is a softmax over the hyperbolic distance of subgraph encoding to the label prototypes, which indicates the probability that the subgraph belongs to the class $y_k$. The loss function $\mathcal{L}(p, y)$ is the cross-entropy loss between ground truth labels $y$ and the class distribution vector $p$.

### 3.4 Meta-Learning

In the previous section, we learned a continuous label encoding that is able to capture inductive biases from the subgraph. In this section, we utilize the optimization-based MAML algorithm [12] to transfer the inductive biases from the support set to the query set. To this end, we sample a batch of tasks, where each task $\mathcal{T}_i = \{S_i, Y_i\}_{i=1}^{\|\mathcal{T}_i\|}$. In the meta-training phase, we first optimize the HNN parameters using the Riemannian stochastic gradient descent (RSGD) [2] on support loss, i.e., for each $\mathcal{T}_i^s \in \mathcal{T}_{train}^s : \theta_j^* \leftarrow exp_{\theta_j^*}^c(-\alpha\nabla\mathcal{L}^s)$, where $\alpha$ is the learning rate of RSGD. Using the updated parameters $\theta_j^*$, we record the evaluation results on the query set, i.e., loss on task $\mathcal{T}_i^q \in \mathcal{T}_{train}^q$ is $\mathcal{L}_i^q$. The above procedure is repeated $\eta$ times post which $\mathcal{L}_i^q$ over the batch of tasks $\mathcal{T}_i^q \in \mathcal{T}_{train}^q$ is accumulated for the meta update $\theta^* \leftarrow exp_{\theta*}^c(-\beta\nabla\sum_i\mathcal{L}_i^q)$. The above steps are repeated with the updated $\theta^*$ and a new batch of tasks till convergence. The final updated parameter set $\theta^* \rightarrow \theta$ is transferred to the meta-testing phase. In meta-testing, the tasks $\mathcal{T}_i^s \in \mathcal{T}_{test}^s$ are used for RSGD parameter updates, i.e., $\mathcal{T}_i^s \in \mathcal{T}_{test}^s : \theta_j \leftarrow exp_{\theta_j}^c(-\alpha\nabla\mathcal{L}^s)$ until convergence. The updated parameters $\theta$ are used for the final evaluation on $\mathcal{T}_i^q \in \mathcal{T}_{test}^q$. Our meta-learning procedure is further detailed in Appendix C and the implementation code with our experiments is available at `https://github.com/Akirato/HGRAM`. Details on implementation and broader impacts are provided in Appendices 3.5 and E, respectively.

### 3.5 Implementation Details

H-GRAM is primarily implemented in Pytorch [26], with geoopt [21] and GraphZoo [35] as support libraries for hyperbolic formulations. Our experiments are conducted on a Nvidia V100 GPU with 16 GB of VRAM. For gradient descent, we employ Riemannian Adam [28] with an initial learning rate of 0.01 and standard $\beta$ values of 0.9 and 0.999. The other hyper-parameters were selected based on the best performance on the validation set ($\mathcal{D}_{val}$) under the given computational constraints. In our experiments, we empirically set $k = 2, d = 32, h = 4$, and $\eta = 10$. We explore the following search space and tune our hyper-parameters for best performance. The number of tasks in each batch are varied among 4, 8, 16, 32, and 64. The learning rate explored for both HNN updates and meta updates are $10^{-2}, 5 \times 10^{-3}, 10^{-3}$ and $5 \times 10^{-4}$. The size of hidden dimensions are selected from among 64, 128, and 256. The final best-performing hyper-parameter setup for real-world datasets is presented in Table 5.

## 4 Experimental Setup

Our experiments aim to evaluate the performance of the proposed H-GRAM model and investigate the following research questions:

**RQ1:** Does our hyperbolic meta-learning algorithm outperform the Euclidean baselines on various meta-learning problems?

**RQ2:** How does our model perform and scale in comparison to other HNN formulations in standard graph problems?

**RQ3:** How does H-GRAM model's performance vary with different few-shot settings, i.e., different values of $k$ and $N$?

**RQ4:** What is the importance of different meta information components?

We use a set of standard benchmark datasets and baseline methods to compare the performance of H-GRAM on meta-learning and graph analysis tasks. The HNN models do not scale to the large datasets used in the meta-learning task, and hence, we limit our tests to Euclidean baselines. To compare against HNN models, we rely on standard node classification and link prediction on small datasets. Also, we do not consider other learning paradigms, such as self-supervised learning because they require an exhaustive set of nodes and labels and do not handle disjoint problem settings.

### 4.1 Datasets

For the task of meta-learning, we utilize the experimental setup from earlier approaches [18]; two synthetic datasets to understand if H-GRAM is able to capture local graph information and five real-world datasets to evaluate our model's performance in a few-shot setting.

- **Synthetic Cycle** [18] contains multiple graphs with cycle as the basis with different topologies (House, Star, Diamond, and Fan) attached to the nodes on the cycle. The classes of the node are defined by their topology.
- **Synthetic BA** [18] uses Barabási-Albert (BA) graph as the basis with different topologies planted within it. The nodes are labeled using spectral clustering over the Graphlet Distribution Vector [27] of each node.
- **ogbn-arxiv** [17] is a large citation graph of papers, where titles are the node features and subject areas are the labels.
- **Tissue-PPI** [16, 43] contains multiple protein-protein interaction (PPI) networks collected from different tissues, gene signatures and ontology functions as features and labels, respectively.
- **FirstMM-DB** [25] is a standard 3D point cloud link prediction dataset.
- **Fold-PPI** [43] is a set of tissue PPI networks, where the node features and labels are the conjoint triad protein descriptor [30] and protein structures, respectively.
- **Tree-of-Life** [44] is a large collection of PPI networks, originating from different species.

## 4.2 Baseline Methods

For comparison with HNNs, we utilize the standard benchmark citation graphs of Cora [29], Pubmed [24], and Citeseer [29]. For the baselines, we select the following methods to understand H-GRAM's performance compared to state-of-the-art models in the tasks of meta-learning and standard graph processing.
- **Meta-Graph** [3], developed for few-shot link prediction over multiple graphs, utilizes VGAE [20] model with additional graph encoding signals.
- **Meta-GNN** [40] is a MAML developed over simple graph convolution (SGC) network [36].
- **FS-GIN** [37] runs Graph Isomorphism Network (GIN) on the entire graph and then uses the few-shot labelled nodes to propagate loss and learn.
- **FS-SGC** [36] is the same as FS-GIN but uses SGC instead of GIN as the GNN network.
- **ProtoNet** [31] learn a metric space over label prototypes to generalize over unseen classes.
- **MAML** [12] is a Model-Agnostic Meta Learning (MAML) method that learns on multiple tasks to adapt the gradients faster on unseen tasks.
- **HMLP** [13], **HGCN** [4], and **HAT** [15] are the hyperbolic variants of Euclidean multi-layer perceptron (MLP), Graph Convolution Network (GCN), and Attention (AT) networks that use hyperbolic gyrovector operations instead of the vector space model.

It should be noted that not all the baselines can be applied to both node classification and link prediction. Hence, we compare our model against the baselines only on the applicable scenarios.

# 5 Experimental Results

We adopt the following standard problem setting which is widely studied in the literature [18]. The details of the datasets used in our experiments are provided in Table 1. In the case of synthetic datasets, we use a 2-way setup for disjoint label problems, and for the shared label problems the cycle graph and Barabási–Albert (BA) graph have 17 and 10 labels, respectively. The evaluation of our model uses 5 and 10 gradient update steps in meta-training and meta-testing, respectively. In the case of real-world datasets, we use 3-shot and 16-shot setup for node classification and link prediction, respectively. For real-world disjoint labels problem, we use the 3-way classification setting. The evaluation of our model uses 20 and 10 gradient update steps in meta-training and meta-testing, respectively. In the case of Tissue-PPI dataset, we perform each 2-way protein function task three times and average it over 10 iterations for the final result. In the case of link prediction task, we need to ensure the distinct nature of support and query set in all meta-training tasks. For this, we hold out a fixed set comprised of 30% and 70% of the edges as a preprocessing step for every graph for the support and query set, respectively.

## 5.1 RQ1: Performance of Meta-Learning

To analyze the meta-learning capability of H-GRAM, we compare it against previous approaches in this area on a standard evaluation setup. We consider two experimental setups inline with previous evaluation in the literature [18]; (i) with synthetic datasets to analyze performance on different problem setups without altering the graph topology, and (ii) with real-world datasets to analyze per-

Table 1: Basic statistics of the datasets used in our experiments. The columns present the dataset task (node classification or link prediction), number of graphs $|\mathcal{G}^{\cup}|$, nodes $|V|$, edges $|E|$, node features $|X|$, and labels $|Y|$. Node, Link, and N/L indicates whether the datasets are used for node classification, link prediction, and both, respectively.

| Dataset | Task | $\mathbf{|\mathcal{G}^{\cup}|}$ | $\mathbf{|V|}$ | $\mathbf{|E|}$ | $\mathbf{|X|}$ | $\mathbf{|Y|}$ |
|---|---|---|---|---|---|---|
| Synth. Cycle | Node | 10 | 11,476 | 19,687 | - | 17 |
| Synth. BA | Node | 10 | 2,000 | 7,647 | - | 10 |
| ogbn-arxiv | Node | 1 | 169,343 | 1,166,243 | 128 | 40 |
| Tissue-PPI | Node | 24 | 51,194 | 1,350,412 | 50 | 10 |
| FirstMM-DB | Link | 41 | 56,468 | 126,024 | 5 | 2 |
| Fold-PPI | Node | 144 | 274,606 | 3,666,563 | 512 | 29 |
| Tree-of-Life | Link | 1,840 | 1,450,633 | 8,762,166 | - | 2 |
| Cora | N/L | 1 | 2,708 | 5,429 | 1,433 | 7 |
| Pubmed | N/L | 1 | 19,717 | 44,338 | 500 | 3 |
| Citeseer | N/L | 1 | 3,312 | 4,732 | 3,703 | 6 |

Table 2: Performance of H-GRAM and the baselines on synthetic and real-world datasets. The top three rows define the task, problem setup (Single Graph (SG), Multiple Graphs (MG), Shared Labels (SL) or Disjoint Labels (DL)) and dataset. The problems with disjoint labels use a 2-way meta-learning setup, and in the case of shared labels, the cycle (S. Cy) and BA (S. BA) graph have 17 and 10 labels, respectively. In our evaluation, we use 5 and 10 gradient update steps in meta-training and meta-testing, respectively. The columns present the average multi-class classification accuracy and 95% confidence interval over five-folds. Note that the baselines are only defined for certain tasks, "-" implies that the baseline is not defined for the task and setup. Meta-Graph is only defined for link prediction. The confidence intervals for the results are provided in Appendix D.

| | Synthetic Datasets | | | | | | Real-world (Node Classification) | | | Real-world (Link Prediction) | |
|---|---|---|---|---|---|---|---|---|---|---|---|
| Task Setup | $\langle SG, DL \rangle$ | | $\langle MG, SL \rangle$ | | $\langle MG, DL \rangle$ | | $\langle SG, DL \rangle$ | $\langle MG, SL \rangle$ | $\langle MG, DL \rangle$ | $\langle MG, SL \rangle$ | $\langle MG, SL \rangle$ |
| Dataset | S. Cy | S. BA | S. Cy | S. BA | S. Cy | S. BA | ogbn-arxiv | Tissue-PPI | Fold-PPI | FirstMM-DB | Tree-of-Life |
| **Meta-Graph** | - | - | - | - | - | - | - | - | - | .719 | .705 |
| **Meta-GNN** | .720 | .694 | - | - | - | - | .273 | - | - | - | - |
| **FS-GIN** | .684 | .749 | - | - | - | - | .336 | - | - | - | - |
| **FS-SGC** | .574 | .715 | - | - | - | - | .347 | - | - | - | - |
| **ProtoNet** | .821 | .858 | .282 | .657 | .749 | .866 | .372 | .546 | .382 | .779 | .697 |
| **MAML** | .842 | .848 | .511 | .726 | .653 | .844 | .389 | .745 | .482 | .758 | .719 |
| **G-META** | .872 | .867 | .542 | .734 | .767 | .867 | .451 | .768 | .561 | .784 | .722 |
| **H-GRAM** | **.883** | **.873** | **.555** | **.746** | **.779** | **.888** | **.472** | **.786** | **.584** | **.804** | **.742** |

formance for practical application. Based on the problem setup, the datasets are partitioned into node-centric subgraphs with corresonding root node's label as the subgraph's ground truth label. The subgraphs are subsequently batched into tasks which are further divided into support set and query set for meta-learning. The evaluation metric for both the tasks of node classification and link prediction is accuracy $\mathcal{A} = |Y = \hat{Y}|/|Y|$. For robust comparison, the metrics are computed over five folds of validation splits in a 2-shot setting for node classification and 32-shot setting for link prediction. Table 2 presents the five-fold average and 95% confidence interval of our experiments on synthetic and real-world datasets, respectively. From the results, we observe that H-GRAM consistently outperforms the baseline methods on a diverse set of datasets and meta-learning problem setups. For the disjoint labels setting, H-GRAM outperforms the best baseline in both the cases of single and multiple graphs. In the case of synthetic graphs, we observe that subgraph methods of H-GRAM and G-Meta outperform the entire graph encoding based approaches showing that subgraph methods are able to limit the over-smoothing problem [6] and improve performance. Also, we observe that meta-learning methods (ProtoNet and MAML) are unreliable in their results producing good results for some tasks and worse for others, whereas H-GRAM is consistently better across the board. Hence, we conclude that using label prototypes to learn inductive biases and transferring them using MAML meta updates is a more robust technique. We note that H-GRAM, unlike previous HNN models, is able to handle graphs with edges and nodes in the order of millions, as evident by the performance on large real-world datasets including ogbn-arxiv, Tissue-PPI, Fold-PPI, and Tree-of-Life. Our experiments clearly demonstrate the significant performance of H-GRAM in a wide-range of applications and prove the effectiveness of meta-learning in HNN models.

Table 3: Comparison with HNN models on standard benchmarks. We compare the Single Graph, Shared Labels (SG,SL) setup of the H-GRAM model to the baselines. The columns report the average multi-class classification accuracy and 95% confidence interval over five-folds on the tasks of node classification (Node) and link prediction (Link) in the standard citation graphs.

| Dataset | Cora | | Pubmed | | Citeseer | |
|---|---|---|---|---|---|---|
| Task | Node | Link | Node | Link | Node | Link |
| HMLP | .754±.029 | .765±.047 | .657±.045 | .848±.038 | .879±.078 | .877±.090 |
| HAT | .796±.036 | **.792±.038** | .681±.034 | .908±.038 | .939±.034 | .922±.036 |
| HGCN | .779±.026 | .789±.030 | .696±.029 | **.914±.031** | **.950±.032** | .928±.030 |
| **H-GRAM** | **.827±.037** | .790±.026 | **.716±.029** | .896±.025 | .924±.033 | **.936±.030** |

## 5.2 RQ2: Comparison with HNN models

The current HNN formulations of HMLP, HAT, and HGCN do not scale to large datasets, and hence, we were not able to compare them against H-GRAM on large-scale datasets. However, it is necessary to compare the standard HNN formulations with H-GRAM to understand the importance of subgraph encoders and meta-learning. Thus, we utilize the single graph and shared labels setup of H-GRAM to evaluate its performance on citation networks of Cora, Pubmed, and Citeseer for both tasks of node classification and link prediction. We also compare the time taken by our model and other HNNs on varying number of nodes $\left(|\mathcal{V}| = \{10^i\}_{i=1}^7\right)$ in the Synthetic BA graph. For this experiment, we also consider a multi-GPU version of H-GRAM that parallelizes the HNN update computations and accumulates them for meta update. From our results, presented in Table 3, we observe that H-GRAM is, on average, able to outperform the best baseline. This shows that our formulation of HNN models using meta-learning over node-centric subgraphs is more effective than the traditional models, while also being scalable over large datasets. The scalability allows for multi-GPU training and translates the performance gains of HNNs to larger datasets. In the results provided in Figure 3, we observe that the time taken by the models is inline with their parameter complexity (HMLP≤HGCN≤HAT≤H-GRAM). However, the traditional HNNs are not able to scale beyond $|\mathcal{V}| = 10^4$, whereas, H-GRAM is able to accommodate large graphs. Another point to note is that H-GRAM(multi) is able to parallelize well over multiple GPUs with its time taken showing stability after $10^4$ nodes (which is the size of nodes that a single GPU can accommodate).

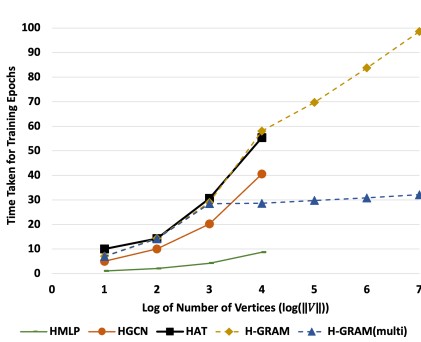

Figure 3: Time taken (per epoch) by H-GRAM compared to other HNNs with varying number of nodes $|\mathcal{V}| = \{10^i\}_{i=1}^7$ in Syn. BA graph. H-GRAM(multi) is the multi-GPU version of H-GRAM.

## 5.3 RQ3: Challenging Few-shot Settings

To understand the effect of different few-shot learning scenarios, we vary the number of few-shots $M$ and hops $K$ in the neighborhood. For the experiment on few-shot classification, we consider the problems of node classification on Fold-PPI dataset and link prediction on FirstMM-DB dataset. We vary $M = 1, 2, 3$ and $M = 16, 32, 64$ for node classification and link prediction, respectively, and calculate the corresponding accuracy and 95% confidence interval of H-GRAM. The results for this experiment are presented in Figures 4a and 4b for node classification and link prediction, respectively. To determine the effect of hops in the neighborhood, we vary $K = 1, 2, 3$ for the same problem setting and compute the corresponding performance of our model. The results for varying neighborhood sizes are reported in Figure 4c. In the results on varying the number of few-shots, we observe a linear trend in both the tasks of node classification and link prediction, i.e., a linear increase in H-GRAM's accuracy with an increase in the number of shots in meta-testing. Thus, we conclude that H-GRAM, like other generic learning models, performs better with an increasing number of training samples. In the experiment on increasing the neighborhood size, we observe that in the task of node classification, $K = 3$ shows the best performance, but in link prediction $K = 2$

has the best performance, with a significant drop in $K = 3$. Thus, for stability, we choose $K = 2$ in our experiments. The trend in link prediction also shows that larger neighborhoods can lead to an increase in noise, which can negatively affect performance.

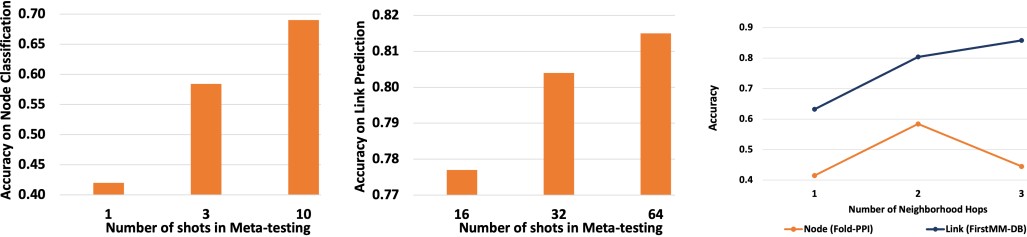

(a) Number of Shots (vs) Accuracy for node classification on Fold-PPI dataset. (b) Number of Shots (vs) Accuracy for link prediction on FirstMM-DB dataset. (c) Number of hops (vs) Accuracy on the task of node classification and link prediction.

Figure 4: Performance of H-GRAM on challenging few-shot settings. The reported accuracies are multi-class classification accuracy averaged over five-fold runs of our model.

## 5.4 RQ4: Ablation Study

In this section, we aim to understand the contribution of the different components in our model. To this end, we compare variants of our model by (i) varying the base HNN model (HMLP, HAT, and HGCN), and (ii) deleting individual meta-learning components - H-ProtoNet implies H-GRAM without meta updates and H-MAML implies H-GRAM without prototypes. The model variants are compared on the real-world datasets and the results are presented in Table 4. The ablation study indicates that meta gradient updates and label prototypes contribute to ≈16% and ≈6% improvement in H-GRAM's performance, respectively. This clearly demonstrates the ability of label prototypes in encoding inductive biases and that of meta gradients in transferring the knowledge from meta-training to the meta-testing phase. Additionally, from our study on different HNN bases for H-GRAM, we note that the HGCN base outperforms the other bases of HMLP and HAT by ≈19% and ≈2%, respectively. Thus, we choose HGCN as the base in our final model.

Table 4: Ablation study results - H-ProtoNet and H-MAML can be considered H-GRAM's model variants without meta updates and label prototypes, respectively. H-GRAM(HMLP) and H-GRAM(HAT) represents the variant of H-GRAM with HMLP and HAT as base, respectively. Our final model, presented in the last row, uses HGCN as the base model. The columns report the average multi-class classification accuracy and 95% confidence interval over five-folds on different tasks.

| Task | Node Classification | | | Link Prediction | |
|---|---|---|---|---|---|
| **Setup** | $\langle SG, DL \rangle$ | $\langle MG, SL \rangle$ | $\langle MG, DL \rangle$ | $\langle MG, SL \rangle$ | $\langle MG, SL \rangle$ |
| **Dataset** | **ogbn-arxiv** | **Tissue-PPI** | **Fold-PPI** | **FirstMM-DB** | **Tree-of-Life** |
| **H-ProtoNet** | .389±.019 | .559±.027 | .398±.023 | .799±.015 | .716±.004 |
| **H-MAML** | .407±.023 | .762±.056 | .502±.046 | .777±.018 | .739±.005 |
| **H-GRAM(HMLP)** | .370±.036 | .537±.044 | .372±.036 | .772±.028 | .688±.019 |
| **H-GRAM(HAT)** | .462±.032 | .777±.028 | .573±.048 | .794±.023 | .732±.021 |
| **H-GRAM(ours)** | **.472±.035** | **.786±.031** | **.584±.044** | **.804±.021** | **.742±.013** |

## 6 Conclusion

In this paper, we introduce H-GRAM, a scalable hyperbolic meta-learning model that is able to learn inductive biases from a support set and adapt to a query set with disjoint nodes, edges, and labels by transferring the knowledge. We have theoretically proven the effectiveness of node-centric subgraph information in HNN models, and used that to formulate a meta-learning model that can scale over large datasets. Our model is able to handle challenging few-shot learning scenarios and also outperform the previous Euclidean baselines in the area of meta-learning. Additionally, unlike previous HNN models, H-GRAM is also able to scale to large graph datasets.

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
