# Appendix

## A    Preliminaries

In this section, we discuss the hyperbolic operations used in HNN formulations and set up the meta-learning problem.

**Hyperbolic operations:** The hyperbolic gyrovector operations required for training neural networks, in a Poincaré ball with curvature $c$, are defined by Möbius addition ($\oplus_c$), Möbius subtraction ($\ominus_c$), exponential map ($\exp_x^c$), logarithmic map ($\log_x^c$), Möbius scalar product ($\odot_c$), and Möbius matrix-vector product ($\otimes_c$).

$$g_x^{\mathbb{H}} := \lambda_x^2 \ g^{\mathbb{E}} \quad \text{where } \lambda_x := \frac{2}{1 - \|x\|^2} \tag{3}$$

$$x \oplus_c y := \frac{\left(1 + 2c\langle x, y\rangle + c\|y\|^2\right) x + \left(1 - c\|x\|^2\right) y}{1 + 2c\langle x, y\rangle + c^2\|x\|^2\|y\|^2} \tag{4}$$

$$x \ominus_c y := x \oplus_c -y \tag{5}$$

$$\exp_x^c(v) := x \oplus_c \left(\tanh\left(\sqrt{c}\frac{\lambda_x^c\|v\|}{2}\right)\frac{v}{\sqrt{c}\|v\|}\right) \tag{6}$$

$$\log_x^c(y) := \frac{2}{\sqrt{c}\lambda_x^c}\tanh^{-1}\left(\sqrt{c}\| - x \oplus_c y\|\right)\frac{-x \oplus_c y}{\| - x \oplus_c y\|} \tag{7}$$

$$r \odot_c x := \exp_0^c(r\log_0^c(x)), \ \forall r \in \mathbb{R}, x \in \mathbb{H}^d \tag{8}$$

$$M \otimes_c x := \frac{1}{\sqrt{c}}\tanh\left(\frac{\|Mx\|}{\|x\|}\tanh^{-1}\left(\sqrt{c}\|x\|\right)\right)\frac{Mx}{\|Mx\|} \tag{9}$$

Here, $:=$ denotes assignment operation for Möbius operations. The operands $x, y \in \mathbb{H}^d$ are hyperbolic gyrovectors. $g^{\mathbb{E}} = \mathbf{I}_n$ is the Euclidean identity metric tensor and $\|x\|$ is the Euclidean norm of $x$. $\lambda_x$ is the conformal factor between the Euclidean and hyperbolic metric tensor [13].

**Meta-Learning setup:** In meta-learning, the dataset consists of multiple tasks $\mathcal{T}_i \in \mathcal{D}$. The dataset is divided into training ($\mathcal{D}_{train}$), test ($\mathcal{D}_{test}$), and validation ($\mathcal{D}_{val}$) sets, and each task $\mathcal{T}_i$ is divided into a support set ($\mathcal{T}_i^s$) and query set ($\mathcal{T}_i^q$) with corresponding labels $Y_s$ and $Y_q$, respectively. The size of query label set is given by $N = |Y_q|$ and the size of support set in meta-testing is given by $K = |\mathcal{T}_i^s \in \mathcal{D}_{test}|$. This particular setup is also known as the N-ways K-shot learning problem. In the meta-training phase of MAML [12], the model trains on tasks $\mathcal{T}_i^s \in D_{train}$ and is evaluated on $\mathcal{T}_i^q \in \mathcal{D}_{train}$ to learn the meta-information. For few-shot evaluation, the model is trained on $\mathcal{T}_i^s \in \mathcal{D}_{test}$ and evaluated on $\mathcal{T}_i^q \in \mathcal{D}_{test}$, where $|\mathcal{T}_i^s| \ll |\mathcal{T}_i^q|$. The goal is to learn the initial parameters $\theta_*$ from $\mathcal{D}_{train}$ such that they can quickly transition to the parameters $\theta$ for new tasks in $\mathcal{D}_{test}$. The hyper-parameters of the meta-learning setup are tuned using $\mathcal{D}_{val}$.

## B    Theorem Proofs

This section provides the theoretical proofs of the theorems presented in our main paper.

### B.1    Proof of Theorem 1

**Theorem.** *For a set of paths $P_{uv}$ between nodes $u$ and $v$, let us define $D_{g\mu}^{p_i}$ as the geometric mean of degree of nodes in a path $p_i \in P_{uv}$, $p_{uv}$ as the shortest path, and $\mathcal{I}_{uv}$ as the influence of node $v$ on $u$. Also, let us say define $D_{g\mu}^{min} = min\left\{D_{g\mu}^{p_i} \forall p_i \in P_{uv}\right\}$, then the relation $\mathcal{I}_{uv} \leq exp_u^c\left(K/\left(D_{g\mu}^{min}\right)^{\|p_{uv}\|}\right)$ (where K is a constant) holds for message propagation in HGCN models.*

*Proof.* The aggregation in HGCN model is defined as:

$$x_u^{\mathbb{H}} = exp_{x_u^{\mathbb{H}}}^c\left(\frac{1}{D_u}\sum_{i \in \mathcal{N}(u)} w_{ui}log_{x_u^{\mathbb{H}}}^c\left(x_i^{\mathbb{H}}\right)\right)$$

where $x_u$, $D_u$, and $\mathcal{N}(u)$ are the embedding, degree, and neighborhood of the root node, respectively. Note that points in the local tangent space follow Euclidean algebra. For simplicity, let us define the Euclidean vector as $x_i = log_{x_u}^c\left(x_i^{\mathbb{H}}\right)$. Simplifying the aggregation function:

$$x_u^{\mathbb{H}} = exp_{x_u^{\mathbb{H}}}^c\left(\frac{1}{D_u}\sum_{i\in\mathcal{N}(u)}w_{ui}x_i\right)$$

Expanding the aggregation function to cover all possible paths from $u$ to its connected nodes.

$$x_u^{\mathbb{H}} = exp_{x_u^{\mathbb{H}}}^c\left(\frac{1}{D_u}\sum_{i\in\mathcal{N}(u)}w_{ui}\frac{1}{D_i}\sum_{j\in\mathcal{N}(i)}w_{ij}...\right.$$
$$\left.\frac{1}{D_m}\sum_{n\in\mathcal{N}(m)}w_{mn}...\frac{1}{D_o}\sum_{o\in\mathcal{N}(k)}w_{ko}x_o\right)$$

The influence of a node $v$ on $u$ is given by:

$$\mathcal{I}_{uv} = exp_0^c\left(\left\|\frac{\partial log_x^c\left(x_u^{\mathbb{H}}\right)}{\partial log_x^c\left(x_v^{\mathbb{H}}\right)}\right\|\right)$$

Simplifying tangent space vectors,

$$\mathcal{I}_{uv} = exp_0^c\left(\left\|\frac{\partial x_u}{\partial x_v}\right\|\right)$$

$$\left\|\frac{\partial x_u}{\partial x_v}\right\| = \left\|\frac{\partial}{\partial x_v}\left(\frac{1}{D_u}\sum_{i\in\mathcal{N}(u)}w_{ui}\frac{1}{D_i}\sum_{j\in\mathcal{N}(i)}w_{ij}...\right.\right.$$
$$\left.\left.\frac{1}{D_m}\sum_{n\in\mathcal{N}(m)}w_{mn}...\frac{1}{D_o}\sum_{o\in\mathcal{N}(k)}w_{ko}x_o\right)\right\|$$

Given that the partial derivative is with respect to $x_v$, only paths between $u$ and $v$ will be non-zero, all other paths shall be zero, i.e.,

$$\left\|\frac{\partial x_u}{\partial x_v}\right\| = \left\|\frac{\partial}{\partial x_v}\left(\frac{1}{D_u}w_{up_i^1}\frac{1}{D_{p_i^1}}w_{p_i^1 p_j^1}...\frac{1}{D_{p_k^1}}w_{p_k^1 v}x_v + ...\right.\right.$$
$$\left.\left.+\frac{1}{D_u}w_{up_i^m}\frac{1}{D_{p_i^m}}w_{p_i^m p_j^m}...\frac{1}{D_{p_k^m}}w_{p_k^m v}x_v\right)\right\|$$

where $(u, p_i^t, p_j^t, ..., p_k^t, v)\forall t\in[1,m]$ are the paths between node $u$ and $v$. Aggregating the terms together and getting the constants out of the derivative,

$$\left\|\frac{\partial x_u}{\partial x_v}\right\| = \left\|\frac{w_{up_i^1}w_{p_i^1 p_j^1}...w_{p_k^1 v}}{D_u D_{p_i^1}...D_{p_k^1}} + ... + \frac{w_{up_i^m}w_{p_i^m p_j^m}...w_{p_k^m v}}{D_u D_{p_i^m}...D_{p_k^m}}\right\|\cdot\left\|\frac{\partial x_v}{\partial x_v}\right\|$$
$$= \left\|\frac{w_{up_i^1}w_{p_i^1 p_j^1}...w_{p_k^1 v}}{D_u D_{p_i^1}...D_{p_k^1}} + ... + \frac{w_{up_i^m}w_{p_i^m p_j^m}...w_{p_k^m v}}{D_u D_{p_i^m}...D_{p_k^m}}\right\|$$
$$\leq \left\|m * max\left(\frac{w_{up_i^1}w_{p_i^1 p_j^1}...w_{p_k^1 v}}{D_u D_{p_i^1}...D_{p_k^1}}, ..., \frac{w_{up_i^m}w_{p_i^m p_j^m}...w_{p_k^m v}}{D_u D_{p_i^m}...D_{p_k^m}}\right)\right\|$$

Let us say,

$$t^* = \arg\max_t\left(\left\{\frac{w_{up_i^t}w_{p_i^t p_j^t}...w_{p_k^t v}}{D_u D_{p_i^1}...D_{p_k^t}}\right\}_{t=1}^{t=m}\right) \tag{10}$$

Then,

$$\left\|\frac{\partial x_u}{\partial x_v}\right\| \leq \left\|m * \frac{w_{up_i^{t*}} w_{p_i^{t*} p_j^{t*}} ... w_{p_k^{t*} v}}{D_u D_{p_i^{t*}} ... D_{p_k^{t*}}}\right\|$$

Aggregating the constants and substituting the geometric mean, we get;

$$\left\|\frac{\partial x_u}{\partial x_v}\right\| \leq K \times \left(\frac{1}{\left(D_u D_{p_i^{t*}} ... D_{p_k^{t*}}\right)^{1/n_*}}\right)^{n_*} = \frac{K}{\left(D_{g\mu}^{t*}\right)^{n_*}}$$

Substituting the variables with shortest paths and minimum degree,

$$\left\|\frac{\partial x_u}{\partial x_v}\right\| \leq \frac{K}{\left(D_{g\mu}^{t*}\right)^{n_*}} \leq \frac{K}{\left(D_{g\mu}^{min}\right)^{\|p_{uv}\|}}$$

With transitive property and adding exponential map on both sides;

$$exp_u^c\left(\left\|\frac{\partial x_u}{\partial x_v}\right\|\right) \leq exp_u^c\left(K/\left(D_{g\mu}^{min}\right)^{\|p_{uv}\|}\right)$$

$$\mathcal{I}_{uv} \leq exp_u^c\left(K/\left(D_{g\mu}^{min}\right)^{\|p_{uv}\|}\right)$$

$\square$

## B.2   Proof of Theorem 2

**Theorem.** *Given the subgraph $S_u$ of graph $\mathcal{G}$ centered at node $u$, with the corresponding label $Y_u$, let us define a node $v \in \mathcal{G}$ with maximum influence on $u$, i.e., $v = \arg\max_t(\{\mathcal{I}_u t, t \in \mathcal{N}(u) \setminus u\})$. For a set of paths $P_{uv}$ between nodes $u$ and $v$, let us define $D_{g\mu}^{p_i}$ as the geometric mean of degree of nodes in a path $p_i \in P_{uv}$, $\|p_{uv}\|$ is the shortest path length, and $D_{g\mu}^{min} = min\{D_{g\mu}^{p_i} \forall p_i \in P_{uv}\}$. Then, the information loss between encoding the $\mathcal{G}$ and $S_u$ is bounded by $\delta_{\mathbb{H}}(\mathcal{G}, S_u) \leq exp_u^c\left(K/\left(D_{g\mu}^{min}\right)^{\|p_{uv}\|+1}\right)$ (where $K$ is a constant).*

*Proof.* The information loss between encoding the entire graph $\mathcal{G}$ and node-centric local subgraph $S_u$ with root node $u$ is given by;

$$\delta_{\mathbb{H}}(\mathcal{G}, S_u) = exp_0^c\left(\delta(\mathcal{G}, S_u)\right)$$

$$\delta(\mathcal{G}, S_u) = log_0^c\left(\mathcal{I}_{\mathcal{G}}(u)\right) - log_0^c\left(\mathcal{I}_{S_u}(u)\right)$$

$$= \left(\left\|\frac{\partial x_u}{\partial x_1}\right\| + ... + \left\|\frac{\partial x_u}{\partial x_n}\right\|\right) - \left(\left\|\frac{\partial x_u}{\partial x_{i_1}}\right\| + ... + \left\|\frac{\partial x_u}{\partial x_{i_m}}\right\|\right)$$

Delete the overlapping nodes in the paths,

$$= \left\|\frac{\partial x_u}{\partial x_{t_1}}\right\| + \left\|\frac{\partial x_u}{\partial x_{t_2}}\right\| + ... + \left\|\frac{\partial x_u}{\partial x_{t_{n-m}}}\right\|$$

Using Theorem 1,

$$\leq \frac{K_{t_1}}{\left(D_{g\mu}^{t_1}\right)^{\|p_{uv}^{t_1}\|}} + \frac{K_{t_2}}{\left(D_{g\mu}^{t_2}\right)^{\|p_{uv}^{t_2}\|}} + ... + \frac{K_{t_{n-m}}}{\left(D_{g\mu}^{t_{n-m}}\right)^{\|p_{uv}^{t_{n-m}}\|}}$$

$$\leq (n-m) \times K_{min}/\left(D_{g\mu}^{min}\right)^{\|p_{uv}^{min}+1\|}$$

$$\leq (n-m) \times K_{min}/\left(D_{g\mu}^{min}\right)^{\|p_{uv}+1\|} = K/\left(D_{g\mu}^{min}\right)^{\|p_{uv}+1\|}$$

$$\delta(\mathcal{G}, S_u) \leq K/\left(D_{g\mu}^{min}\right)^{\|p_{uv}+1\|}$$

$$exp_0^c\left(\delta(\mathcal{G}, S_u)\right) \leq exp_0^c\left(K/\left(D_{g\mu}^{min}\right)^{\|p_{uv}+1\|}\right)$$

$$\delta_{\mathbb{H}}(\mathcal{G}, S_u) \leq exp_u^c\left(K/\left(D_{g\mu}^{min}\right)^{\|p_{uv}\|+1}\right)$$

$\square$

Table 5: Hyper-parameter setup of real-world datasets. The columns present the number of tasks in each batch (# Tasks), HNN update learning rate ($\alpha$), meta update learning rate ($\beta$), and size of hidden dimensions (d).

| Dataset | # Tasks | $\alpha$ | $\beta$ | d |
|---|---|---|---|---|
| **arxiv-ogbn** | 32 | $10^{-2}$ | $10^{-3}$ | 256 |
| **Tissue-PPI** | 4 | $10^{-2}$ | $5 \times 10^{-3}$ | 128 |
| **Fold-PPI** | 16 | $5 \times 10^{-3}$ | $10^{-3}$ | 128 |
| **FirstMM-DB** | 8 | $10^{-2}$ | $5 \times 10^{-4}$ | 128 |
| **Tree-of-Life** | 8 | $5 \times 10^{-3}$ | $5 \times 10^{-4}$ | 256 |

# C   H-GRAM Algorithm

Algorithm 1 provides the details of the procedure of the H-GRAM model.

---

**Algorithm 1:** H-GRAM meta-learning algorithm

---

**Input:** Graphs $\mathcal{G}^{\cup} = \{\mathcal{G}_i\}_{i=1}^{\|\mathcal{G}^{\cup}\|}$, Ground truth $Y$;
**Output:** Predictor $P_\theta$, parameters $\theta$;
1 Initialize $\theta_*$ and $\theta$ as HGCN model and meta-update parameters, respectively ;
2 # Partition graphs into node-centric subgraphs
3 $S_1, S_2, ...S_{\|V\|} = Partition\,(\mathcal{G}^{\cup})$ ;
4 # Batch graphs into tasks for meta-learning
5 $\mathcal{T} = \{\mathcal{T}_1, \mathcal{T}_2, ..., \mathcal{T}_{\|\mathcal{T}\|}\}$;
6 **while** *not converged* **do**
7    $\mathcal{T}_{train} \leftarrow sample(\mathcal{T})$;
8    **for** $\mathcal{T}_i \in \mathcal{T}_{train}$ **do**
9      # Batch of support and query set from the tasks
10      $\{S_u\}^s, \{Y_u\}^s \leftarrow \mathcal{T}_i^s$;
11      $\{S_u\}^q, \{Y_u\}^q \leftarrow \mathcal{T}_i^q$;
12      **for** $j \in [1, \eta]$ **do**
13        # Update $\theta^*$ using support set
14        $h_u^s = HGCN_{\theta_{j-1}^*}(S_u{}^s)$
15        $e_u^s = \frac{\sum_{i=1}^{\|V\|} \gamma_{iu} h_{iu}^s}{\sum_{i=1}^{\|V\|} \gamma_{iu}}$
16        $c_k^s = \frac{\sum_{Y_u = y_k} \gamma_i e_i^s}{\sum_{Y_u = y_k} \gamma_i}$
17        $p_k^s = \frac{e^{(-d_{\mathbb{H}}^c(e_u^s, c_k^s))}}{\sum_k e^{(-d_{\mathbb{H}}^c(e_u^s, c_k^s))}}$
18        $\mathcal{L}^s = \mathcal{L}(p^s, Y_u^s) = \sum_j y_i^s \log p_j^s$
19        $\theta_j^* \leftarrow exp_{\theta_{j-1}^*}^c(-\alpha \nabla \mathcal{L}^s)$
20        # Record evaluation with $\theta^*$ on query set
21        $h_u^q = HGCN_{\theta_j^*}(S_u{}^q)$
22        $e_u^q = \frac{\sum_{i=1}^{\|V\|} \gamma_{iu} h_{iu}^q}{\sum_{i=1}^{\|V\|} \gamma_{iu}}$
23        $c_k^q = \frac{\sum_{Y_u = y_k} \gamma_i e_i^q}{\sum_{Y_u = y_k} \gamma_i}$
24        $p_k^q = \frac{e^{(-d_{\mathbb{H}}^c(e_u^q, c_k^q))}}{\sum_k e^{(-d_{\mathbb{H}}^c(e_u^q, c_k^q))}}$
25        $\mathcal{L}_{ij}^q = \mathcal{L}(p^q, Y_u^q) = \sum_j y_i^q \log p_j^q$
26      **end**
27    **end**
28    # Update meta-learning parameter $\theta$
29    $\theta \leftarrow exp_\theta^c(-\beta \nabla \sum_i \mathcal{L}_{iu}^q)$
30 **end**

---

Table 6: Performance of H-GRAM and the baselines on synthetic and real-world datasets. The top three rows define the task, problem setup (Single Graph (SG), Multiple Graphs (MG), Shared Labels (SL) or Disjoint Labels (DL)) and dataset. The problems with disjoint labels use a 2-way meta-learning setup, and in the case of shared labels, the cycle and BA graph have 17 and 10 labels, respectively. In our evaluation, we use 5 and 10 gradient update steps in meta-training and meta-testing, respectively. The columns present the average multi-class classification accuracy and 95% confidence interval over five-folds. Note that the baselines are only defined for certain tasks, "-" implies that the baseline is not defined for the task and setup. Meta-Graph is only defined for link prediction.

| Task | Node Classification | | Node Classification | | Node Classification | | Node Classification | | | Link Prediction | |
|---|---|---|---|---|---|---|---|---|---|---|---|
| Setup | $\langle SG, DL \rangle$ | | $\langle MG, SL \rangle$ | | $\langle MG, DL \rangle$ | | $\langle SG, DL \rangle$ | $\langle MG, SL \rangle$ | $\langle MG, DL \rangle$ | $\langle MG, SL \rangle$ | $\langle MG, SL \rangle$ |
| Dataset | Syn. Cycle | Syn. BA | Syn. Cycle | Syn. BA | Syn. Cycle | Syn. BA | ogbn-arxiv | Tissue-PPI | Fold-PPI | FirstMM-DB | Tree-of-Life |
| Meta-Graph | - | - | - | - | - | - | - | - | - | .719±.018 | .705±.004 |
| Meta-GNN | .720±.191 | .694±.098 | - | - | - | - | .273±.107 | - | - | - | - |
| FS-GIN | .684±.126 | .749±.093 | - | - | - | - | .336±.037 | - | - | - | - |
| FS-SGC | .574±.081 | .715±.088 | - | - | - | - | .347±.004 | - | - | - | - |
| ProtoNet | .821±.173 | .858±.126 | .282±.039 | .657±.030 | .749±.160 | .866±.186 | .372±.015 | .546±.022 | .382±.027 | .779±.018 | .697±.009 |
| MAML | .842±.181 | .848±.186 | .511±.044 | .726±.020 | .653±.082 | .844±.177 | .389±.018 | .745±.045 | .482±.054 | .758±.022 | .719±.011 |
| G-META | .872±.113 | .867±.129 | .542±.039 | .734±.033 | .767±.156 | .867±.183 | .451±.028 | .768±.025 | .561±.052 | .784±.025 | .722±.028 |
| **H-GRAM** | **.883±.145** | **.873±.120** | **.555±.041** | **.746±.028** | **.779±.132** | **.888±.182** | **.472±.035** | **.786±.031** | **.584±.044** | **.804±.021** | **.742±.013** |

# D  Confidence Intervals

In the results presented in Table 6, the reported mean demonstrates the predictive power of H-GRAM and the 95% confidence interval estimates the degree of uncertainty. The large confidence interval in the results on synthetic datasets is because in meta-testing, we only sampled two-labels in each fold. In some cases where the structure of the local subgraphs in meta-training is significantly different compared to meta-testing, our model has poor performance due to limited scope of knowledge transfer. We observe that, in the limited number of data split possibilities in synthetic datasets, there generally is a case where our model does not perform well which results in a larger confidence interval. Real-world datasets contain many more labels, and hence, we are able to sample more for meta-testing, e.g., 5 labels for 3-way classification. This reduces the possibility of atypical results, thus leading to smaller intervals.

# E  Broader Impacts

Our model has the potential to impact various applications that involve graph-structured data, such as social network analysis, bioinformatics, and recommendation systems. Furthermore, the ability to generalize information from subgraph partitions of large datasets can be especially beneficial for applications with limited labeled data, such as in the fields of healthcare and finance. Moreover, H-GRAM also addresses several challenges in HNNs, including inductive learning, over-smoothing, and few-shot learning. These capabilities can be used to improve the performance of HNNs in various tasks such as node classification, link prediction, and graph classification. However, it is important to note that this model also has certain limitations. In particular, H-GRAM is based on a specific type of hyperbolic space, which may not be applicable to certain types of graph-structured data, and there are some assumptions made in the proof of our theoretical results which may not hold in general. Additionally, the meta-learning setup may not be suitable for all types of tasks, and further research is needed to test the performance of H-GRAM on other types of tasks. As a future direction, it would be interesting to investigate the effect of inadequate local subgraphs, scalability of H-GRAM on even larger datasets and explore the effectiveness of H-GRAM on other types of tasks with temporal or multi-modal graph data.