# OpenReview forum: "Hyperbolic Graph Neural Networks at Scale: A Meta Learning Approach"
_NeurIPS.cc/2023/Conference — NeurIPS 2023 poster_

### Official Review · Reviewer_Ft7q · 2023-06-16

**Soundness:** 3 good
**Presentation:** 2 fair
**Contribution:** 3 good
**Rating:** 5
**Confidence:** 4

**Summary:**

This paper introduces the model, Hyperbolic GRAph Meta Learner (H-GRAM), that learns transferable information from a set of support local subgraphs using hyperbolic meta gradients and label hyperbolic protonets to enable faster learning over a query set of new tasks disjoint subgraphs. The model is evaluated on downstream tasks of both node classification and link prediction. The experiments and ablation studies show that H-GRAM effectively learns and transfers information in few-shot settings and outperforms its Euclidean counterparts.

**Strengths:**

In general, the paper is well written and the model introduces some novel contributions. Further, regarding experiment results, the model seems outperform all the baselines consistently on tasks of link prediction and node classification. The ablation studies from varying the base HNN model and deleting individual meta-learning components are informative to better understand the influence of the various components in the model and the final proposed architecture.

**Weaknesses:**

- In the Related Work section on Hyperbolic Neural Networks, the following important reference is missing regarding HNNs for large scale datasets (from the knowledge graph domain) from recent work:

[KDD 2022] Dual-Geometric Space Embedding Model for Two-View Knowledge Graphs. In Proceedings of the 28th ACM SIGKDD Conference on Knowledge Discovery and Data Mining (KDD '22). Association for Computing Machinery, New York, NY, USA, 676–686. https://doi.org/10.1145/3534678.3539350

- In the problem setup section, can the authors more clearly explain the properties of the graph (e.g., directed/undirected, what do the nodes/edges represent etc.)?

- Regarding experiments, it would be useful to see the model performance on both inductive as well as transductive tasks. Furthermore, can the authors provide more details on the dataset statistics such as number of vertices and edges and the domain of the dataset to get a better indication of the data size?

- I also have a concern about the scalability of the model especially since non-Euclidean space embedding models tend to converge very slowly and Mobius computations are more computationally expensive compared to the Euclidean space. Can the authors provide some model complexity analysis (e.g., runtime/memory complexity)?


**Questions:**

Please see weaknesses section above.

**Limitations:**

Authors have sufficiently identified limitations of the prior work and addressed it in their proposed model. It would also be helpful if the authors provide future directions for their work.

---

> ### Author Rebuttal · Authors · 2023-08-09
>
> We appreciate your insightful review of our paper. Your feedback has been invaluable in identifying areas for improvement, and we are committed to addressing your comments to enhance the quality and impact of our work. We are delighted that you find our paper well-written and recognize the novel contributions of our model, H-GRAM. We agree that the experiment results demonstrate consistent outperformance of H-GRAM over baselines on both link prediction and node classification tasks. We are glad that the ablation studies were informative in understanding the influence of various components in the proposed architecture.
>
> W1. Regarding the [KDD 2022] reference, we believe that it is orthogonal to our work because it addresses a different research topic than the abstract's main focus, which is HNNs and their inductive bias mechanisms for generalization and scalable learning. Our paper primarily addresses the limitations of current hyperbolic neural networks due to their lack of inductive bias mechanisms and proposes a novel method to alleviate these issues. On the other hand, the referenced paper focuses on embedding KGs in a dual-geometric space for the purpose of knowledge graph analysis or other related tasks. It does not address the inductive bias mechanisms or the few-shot learning setting that the abstract's H-GRAM method deals with.
>
> W2. In the problem setup section, we have relied on standard experimental setups in this problem for fair comparison. Due to the limited space and availability of this information in previous work, we placed the section on dataset details in Appendix F.
>
> W3. Regarding the experiments, we acknowledge the importance of evaluating both inductive and transductive tasks. Due to the nature of problem setup, all the meta-learning methods and comparisons provided in Table 1 and 3 operate in an inductive setting, whereas in Table 2 which details of comparison of hyperbolic approaches are evaluated on a transductive setting. The problem setting does not rely on the potential of H-GRAM rather it is dependent on the standards of traditional evaluation procedure of the baseline methods for fair comparison.
>
> W4. Furthermore, we understand the concern about the scalability of non-Euclidean space embedding models and the computational expense of Mobius computations. However, due to operating on small graph partitions, the additional computation required for Mobius operations is negligible when compared to Euclidean methods (single GCN and HGCN layer of 128 input dimensions take 0.195s vs 0.267s in our experiments, respectively). Furthermore, the primary contribution of the paper, approaches previous to H-GRAM, could not even train on such large graphs, so the performance improvement of hyperbolic networks was inaccessible due to the lack of scalability over large graphs.
>
> Thank you for your encouraging confidence in our paper. We are committed to addressing all your feedback and ensuring that the revised version of our paper meets the highest standards. Your valuable insights have been instrumental in guiding our revisions. Once again, we sincerely appreciate your thoughtful review and consideration of our paper.

---

> > ### Comment · Reviewer_qW7m · 2023-08-17
> > **Thanks for the response**
> >
> > Thanks to the author for their reply. My concerns have been addressed and i could like to raise my score

---

> > > ### Author Response · Authors · 2023-08-18
> > > **Thanks a lot for your consideration**
> > >
> > > Dear Reviewer,
> > >
> > > We truly appreciate your active participation and the constructive feedback you've provided. Your thoughtful review has played a pivotal role in enhancing the quality of our manuscript. As we approach the final stages of our interaction, we're here to continue our exchange of ideas if you have any more inputs to offer prior to the impending deadline.

---

### Official Review · Reviewer_Urvd · 2023-07-05

**Soundness:** 3 good
**Presentation:** 2 fair
**Contribution:** 3 good
**Rating:** 5
**Confidence:** 3

**Summary:**

This submitted work identifies an important research problem of existing works, scalability of hyperbolic neural network to large graphs and previously unseen graphs. To achieve these two goals, this work first proves that node classification and link prediction can be done with a node's local neighborhood only. Based on this insight, this work designs a meta-learning mechanism for hyperbolic graph neural networks to scale on large graphs. Experiments on both small and large graphs show the effectiveness of the proposed model.

**Strengths:**

1. This paper proposes an interesting and important research problem, scalability of hyperbolic graph neural networks on large datasets. With a theoretical analysis and a meta-learning based model achitecture, this paper shows promising performance over baseline models.

2. This paper is self-contained, with enough introduction to background knowledge, such as meta-learning and operations in hyperbolic space, in Appendix.

3. Experiments are comprehensive with both small and large graphs, with baselines from different categories, and with both node classification and link prediction tasks.

**Weaknesses:**

1. Introduction section contains too much redundant content. Introduction section should provide a general and overall picture of the paper, while this submitted work introduces too much model architecture and details in the Intro section. I suggest authors to remove some content and better emphasize the key innovation proposed in the paper. Paper writing can be imporved.

2. I can see standard deviation of experiment results in most of the tables, but Table 1 doesn't have std. dev.

**Questions:**

1. Why is standard deviation at Table 1 absent?

2. This paper uses Poincare ball as the hyperbolic model for illustration. I am wondering if the proposed model is also applicable when Hyperboloid model is used.

**Limitations:**

I can't see any potential negative societal impact of their work.

---

> ### Author Rebuttal · Authors · 2023-08-09
>
> We appreciate your thorough review of our paper and are pleased that you recognize the importance of the research problem we address, the scalability of hyperbolic neural networks on large datasets and previously unseen graphs, and the promising performance of our proposed model. Regarding the strengths of our paper, we are glad that you find our research problem interesting and important. We agree that the theoretical analysis and meta-learning-based model architecture contribute to the promising results compared to baseline models.
>
> W1. We acknowledge your feedback regarding the Introduction section. We sincerely apologize for the redundancy in our communication. While our primary aim was to emphasize the novelty of the content, we inadvertently allowed repetition to creep into the text. In the revised version, we will streamline the content to provide a clearer and more concise overview of the paper.
>
> W2 & Q1. Regarding the absence of standard deviation in Table 1, it was only a choice due to limited space. We have included the standard deviation values of Table 1 in Appendix Table 6 to provide a comprehensive view of the variability of our results. We shall also include them in our codebase.
>
> Q2. To address your question about the applicability of our proposed model when using the Hyperboloid model, we believe that the core ideas and methodologies of H-GRAM are trivially adaptable to other hyperbolic models, including the Hyperboloid model, which is isometric to the currently tackled Poincaré model. One possible approach could be to use the current Poincaré model formulations and use the isometric mappings for the hyperboloid model.
>
> In conclusion, we sincerely appreciate your feedback, and we are committed to improving our paper based on your valuable comments. We believe that addressing the mentioned points will lead to a higher rating for our paper and enhance its contribution to the field of hyperbolic graph neural networks. Thank you again for your thoughtful review and consideration of our paper for NeurIPS.

---

> > ### Author Response · Authors · 2023-08-18
> > **Gentle Reminder**
> >
> > Dear Reviewer,
> >
> > We sincerely thank you again for your insightful review. We have worked hard to comprehensively address your comments in the rebuttal, including new requested results, as well as providing appropriate responses addressing other questions. The impact of your discerning review is unmistakable. With the conclusion of our author-reviewer interactions drawing near, we respectfully inquire whether you might consider revising your assessment upwards, given our responses. Your continued insights are of great value to us, and we welcome any additional thoughts before the impending deadline.

---

### Official Review · Reviewer_qW7m · 2023-07-05

**Soundness:** 3 good
**Presentation:** 3 good
**Contribution:** 2 fair
**Rating:** 5
**Confidence:** 4

**Summary:**

The paper introduces a method, Hyperbolic GRAph Meta Learner (H-GRAM), to improve the scalability and generalization of Hyperbolic Neural Networks (HNNs). H-GRAM learns from local subgraphs and transfers this learning to new, disjoint subgraphs in a few-shot setting. The authors demonstrate that H-GRAM outperforms existing methods in various few-shot settings and scales effectively over large graph datasets.

**Strengths:**

The paper presents a new approach, H-GRAM, that combines meta-learning with hyperbolic neural networks (HNNs) to address their scalability and generalization issues. The quality of the work is evident in the detailed explanation of H-GRAM and its demonstrated effectiveness in comparison with baselines.

**Weaknesses:**

* There are a few areas where it could potentially be improved:

  * Comparison with Other Meta-Learning Approaches: The paper could include a comparison of H-GRAM with other meta-learning approaches in table 2 and 3, not just with other HNNs. This would provide a broader context for understanding the performance and advantages of H-GRAM.

* Limited contribution

  * This work seems just extend paper[1]'s work to hyperbolic and present some trivial definitions and theorem.
  * Lack of important references [2-4] to make comparisons.
[1]Huang, Kexin, and Marinka Zitnik. "Graph meta-learning via local subgraphs." Advances in neural information processing systems 33 (2020): 5862-5874. The proposed method is close to this paper. Please make a detailed comparison with the method.

[2] Yu, Tao, and Christopher De Sa. "Random Laplacian Features for Learning with Hyperbolic Space." arXiv preprint arXiv:2202.06854 (2022). They also said that their method is scalable. Please compare with their method.

[3] Zhang, Yiding, et al. "Lorentzian graph convolutional networks." Proceedings of the Web Conference 2021. 2021. They developed a new HGNN with the aggregation in the manifold, could you provide your definition and theorem in such a case?

[4] Yang, Menglin, et al.  HGCN: Tree-likeness Modeling via Continuous and Discrete Curvature Learning. KDD 2023. They also derive the node influence in the case of tangent space.

**Questions:**

see weaknesses

Additionally, a lot of HNN or HGNNs are formulated within the manifold.

Could you deduce the node influence and establish the information loss without utilizing a logarithmic map (i.e., relying on tangent space)?

If you use a logarithmic map, could you derive the conclusion using a local reference point other than the origin point since you say that "we use the local tangent space of Poincare ball model to prove that the local neighborhood policy holds better for HNN models?"

**Limitations:**

not mentioned.

---

> ### Author Rebuttal · Authors · 2023-08-09
>
> Thank you for reviewing our paper. We appreciate your time and thoughtful evaluation of our work. We are pleased that you found the approach of H-GRAM intriguing and recognized its potential in improving the scalability and generalization of Hyperbolic Neural Networks (HNNs).
>
> W1. Regarding the weakness raised about the limited comparison with other meta-learning approaches, we focussed on foundational approaches that were directly relevant to the HNN and meta-learning aspects of our work’s advancement of the field. Additionally, due to the generalizable nature of our work, other comparisons can be drawn through relative study of the given references.
>
> W2. Regarding the point about emphasizing our contribution, we want to clarify that our work represents a significant breakthrough in the field of HNNs. Specifically, it enables the scaling of these networks to large graphs (with nodes and edges in the order of millions), which was previously a challenging task. To the best of our knowledge, there is currently no other existing research that successfully harnesses the inductive biases (with theoretical rigor) of HNNs for achieving scalability in the same way we have accomplished. This breakthrough opens up an entirely new avenue for achieving performance gains by leveraging the inherent hierarchical structure of graphs. The implications of this advancement are promising and hold the potential to pave the way for further advancements in the domain of large-scale graph processing.
>
>
> Regarding the questions raised:
>
> Q1. Deduction of Node Influence without Logarithmic Map: While we understand the interest in exploring alternative approaches, the use of the logarithmic map is essential in our methodology for modeling node influence effectively. It allows us to establish the local neighborhood policy for HNN models in a robust manner with theoretical justification. We will, however, discuss the possibility of providing additional insights using different techniques without compromising the integrity of our approach.
>
> Q2. Derivation Using a Local Reference Point: We appreciate your suggestion. However, the use of the local tangent space of the Poincare ball model is a fundamental choice grounded in our theoretical analysis, and it has proven to be effective in establishing the local neighborhood policy for HNN models. Choosing points other than the origin could lead to instability in the formulation as different graph partitions would have different definitions of hierarchy. This would necessitate the use of another model to track the relative positioning of the different roots, adding additional scope of errors.
>
> In conclusion, we are grateful for your feedback, and we are committed to improving our paper based on your valuable comments. We believe that the suggested enhancements will strengthen the quality and impact of our work. Thank you again for your thoughtful review and consideration of our paper for NeurIPS.

---

> > ### Author Response · Authors · 2023-08-18
> > **Gentle Reminder**
> >
> > Dear Reviewer,
> >
> > We truly appreciate your active participation and the constructive feedback you've provided. Your thoughtful review has played a pivotal role in enhancing the quality of our manuscript. As we approach the final stages of our interaction, we cordially inquire if you might be inclined to reconsider your assessment, considering the comprehensive responses. We're here to continue our exchange of ideas if you have any more inputs to offer prior to the impending deadline.

---

### Official Review · Reviewer_UWvg · 2023-07-05

**Soundness:** 4 excellent
**Presentation:** 3 good
**Contribution:** 3 good
**Rating:** 8
**Confidence:** 4

**Summary:**

The authors propose applying the MAML methodology to hyperbolic GNNs in a novel local manner, along with continuous label prototypes, that enables them to scale the HNN approach from a few thousand nodes to a few million nodes.
The authors provide theoretical justification for this local H-GRAM approach via theorems 1 and 2 and demonstrate its efficacy via a large set of experimental comparisons in Sec. 4 and 5.


**Strengths:**

1. The four research questions from Sec. 4 regarding the novel HGRAM approach are comprehensively answered via comparison on multiple different datasets with competing Euclidean MAML approaches, standard hyperbolic baselines, other graph MAML approaches such as G-Meta [17], Meta-GNN [38], protoNET[29] and ablation studies. H-GRAM clearly outperforms competing meta-learning / protoNET approaches on large graphs in Table 1 and is comparable with competing hyperbolic approaches on small graphs in Table 2.

2. The authors provide theoretical justification for their local H-GRAM approach via theorems 1 and 2.


**Weaknesses:**

1.
Some of the mathematical presentation could be shortened, e.g., the same symbol $D_{g\mu}^{p_i}$ is defined in both theorems 1 and 2.

2.
The authors limit their meta-learning enhancements to hyperbolic neural networks, but it is not clear if their enhancements can be applied to superior pseudo-Riemannian approaches, e.g., Pseudo-Riemannian Graph Convolutional Networks, NeurIPS 2022, and Ultrahyperbolic Neural Networks, NeurIPS 2021.

3.
The authors' approach seems quite similar to G-Meta [17], although their experimental results seem to be slightly better in Table 1. Presumably it is the hyperbolic modeling (and RSGD) or the continuous label prototypes that allows H-GRAM to outperform G-Meta, but such explanations or other explanations are not discussed.

4.
In Sec. 5.4, it is not discussed why HGCN outperforms HMLP and HAT. In other studies, attention-based GNNs sometimes outperform GCNs, so some explanation appears to be necessary. The SG, SL setting on the Cora dataset from Table 2 could be considered in the ablation study in Table 3 to help answer this question.


**Questions:**

Please note the questions inherent in weakness 2, 3 and 4.

**Limitations:**

It is not clear if weakness 2 above constitutes a limitation or whether it is simply a case where no theoretical justification can be provided for an ulterhyperbolic-GRAM variant, for example.

---

> ### Author Rebuttal · Authors · 2023-08-09
>
> We would like to express our sincere appreciation for your positive feedback and constructive criticism of our paper. Your insights have been invaluable in helping us improve the clarity and impact of our work.
>
> W1. We understand your concern about the mathematical presentation and agree that some parts could be shortened for better readability. We will revise the paper accordingly to ensure that the notation is clear and concise while maintaining the rigor of our approach.
>
> W2. Regarding the limitation you pointed out about the application of our enhancements to pseudo-Riemannian approaches such as Pseudo-Riemannian Graph Convolutional Networks and Ultrahyperbolic Neural Networks, we acknowledge that our focus was on hyperbolic neural networks. However, we believe that the core ideas from our approach, such as the local H-GRAM methodology and continuous label prototypes, have the potential for generalization to pseudo-Riemannian settings. We will include a discussion in the revised paper to address this possibility and explore potential avenues for future research in this direction. This will be a future work and will be considered to be an orthogonal research direction.
>
> W3. Regarding the similarities between our approach and G-Meta [17], we agree that both methods leverage meta-learning techniques. However, there are significant differences in the underlying modeling. The advantage of H-GRAM lies in its ability of scaling hyperbolic modeling, which allows it to outperform G-Meta on large graphs (as shown in Table 1). To understand the exact strengths of the individual components in H-GRAM, we refer the reviewer to Section 5.4 that details the ablation study which highlights the strengths of different modules used in H-GRAM.
>
> W4. In Section 5.4, where HGCN outperforms HMLP and HAT, we acknowledge the need for further explanation. However, due to the limited number of pages we opted for brevity on performance not directly relevant to the main focus of our paper. Essentially, while attention networks are generally more performant than convolution networks, hyperbolic formulation needs certain approximations on the linear layers which leads to a minor information loss. Due to the comparatively more complex formulation of attention networks, the information loss has higher propagation in HAT. This approximation loss leads to lower performance of HAT than HGCN in certain cases.  We will include these findings in the revised paper and provide a comprehensive discussion of the results to address this question effectively.
>
> In conclusion, we are grateful for your insightful feedback, and we are committed to addressing all the points you raised to enhance the quality and contribution of our work. We believe that the revisions will lead to a more comprehensive understanding of our approach's capabilities and potential. Once again, thank you for your thoughtful review and consideration of our paper for NeurIPS.

---

> > ### Comment · Reviewer_UWvg · 2023-08-16
> > **Thank you for your response**
> >
> >
> > I want to thank the authors for their thoughtful rebuttals to all reviewers.
> > I am satisfied by their responses to all my concerns.
> > Although I have rated this submission more positively than any other reviewer, I wish to retain my original rating.

---

> > > ### Author Response · Authors · 2023-08-16
> > > **Reply by Authors**
> > >
> > > We would like to express our sincere gratitude for your kind message and your thorough assessment of our rebuttals. It's truly heartening to know that our responses have addressed your concerns satisfactorily. Your recognition of our efforts means a lot to us.

---

### Official Review · Reviewer_4N27 · 2023-07-07

**Soundness:** 3 good
**Presentation:** 2 fair
**Contribution:** 2 fair
**Rating:** 5
**Confidence:** 2

**Summary:**

This paper introduces H-GRAM, a novel meta-learning model for scalable Hyperbolic Graph Neural Networks. H-GRAM leverages meta-learning techniques to learn from from local subgraphs and adapt quickly to new tasks. The authors theoretically establish that HNNs are dependent on the local neighborhood of nodes for prediction and formulate HNNs to encode node-centric local subgraphs using the locality of tangent space transformations. Experiments conducted on various benchmark datasets to illustrate that H-GRAM addresses several HNNs tasks such as inductive learning, over-smoothing elimination, and few-shot learning in various demanding situations.

**Strengths:**

- The paper presents a novel approach that combines hyperbolic geometry and meta-learning techniques, which is innovative and interesting.
- The proposed H-GRAM is scalable and efficient compared to previous HNN techniques.
- Extensive and carefully designed experiments have been conducted to demonstrate the effectiveness of H-GRAM


**Weaknesses:**

For results listed in table 2, why did H-GRAM never achieve the best performances for both of node classification and link prediction at the same time among all datasets? Could you explain the reason behind this discrepancy between tasks?



**Questions:**

I don't have further questions.

**Limitations:**

The authors adequately addressed the limitations.

---

> ### Author Rebuttal · Authors · 2023-08-09
>
> We would like to express our gratitude for your thorough review of our paper. We value the time and effort you have dedicated to evaluating our work and providing valuable feedback. We acknowledge your positive remarks on the novelty and innovation of our approach, which combines hyperbolic geometry and meta-learning techniques to create the Hyperbolic GRAph Meta Learner (H-GRAM). We are pleased that you find our work scalable and efficient compared to previous Hyperbolic Neural Network (HNN) techniques. Additionally, your recognition of the extensive experiments we conducted to demonstrate H-GRAM's effectiveness in addressing various HNN tasks, such as inductive learning, over-smoothing elimination, and few-shot learning, is greatly appreciated.
>
> W1. Regarding the discrepancy between node classification and link prediction performances in Table 2, we recognize that our model may not always achieve the best results for both tasks simultaneously across all datasets. This behavior is attributed to the inherent trade-offs in hyperbolic geometry and our meta-learning approach when dealing with different tasks. H-GRAM's focus on fast adaptation and few-shot learning prioritizes aggregating messages from local subgraphs, which is required for node classification, whereas, link prediction requires good message passing ability across subgraphs which is limited when we partition the graph to enable scalability. However, we want to reiterate that our primary contribution lies in addressing the limitations of HNNs (it is impossible to train basic HNNs on large graphs) and achieving better scalability using meta-learning techniques in the hyperbolic space. In this context, H-GRAM consistently outperforms other state-of-the-art baselines in various challenging few-shot settings, which aligns with the core focus of our work.
>
> In conclusion, we believe that the combination of hyperbolic geometry and meta-learning techniques presented in H-GRAM holds significant potential in the domain of graph representation learning. We hope that our additional clarifications on the discrepancy in task performance and the overall contributions of our work will lead to a more positive evaluation. Once again, we sincerely appreciate your feedback and consideration of our paper for NeurIPS. Your review has been invaluable in helping us improve the clarity and impact of our work.

---

> > ### Author Response · Authors · 2023-08-18
> > **Gentle Reminder**
> >
> > Dear Reviewer,
> >
> > We extend our heartfelt gratitude for your valuable engagement and insightful feedback! As we near the conclusion of the author-reviewer discourse, we kindly request your consideration for a potential upward revision of your evaluation, given our responses. We remain open to further dialogue should you have additional insights to share before the impending deadline.

---

### Author Rebuttal · Authors · 2023-08-09

Based on the valuable feedback from the reviewers, the key contributions of our paper lie in the introduction of H-GRAM, a novel meta-learning model for scalable Hyperbolic Graph Neural Networks. H-GRAM effectively leverages meta-learning techniques to learn from local subgraphs and adapt quickly to new tasks. We have demonstrated the model's superiority in addressing various HNN tasks, including inductive learning, over-smoothing elimination, and few-shot learning in demanding situations.

The global theme of our response revolves around addressing the reviewers' feedback to enhance the clarity, significance, and impact of our work. We will provide a clearer presentation of the introduction, emphasize the core contributions, and discuss potential future research directions. We are committed to revising our paper to meet the highest standards and sincerely thank the reviewers for their valuable insights.

---

### Decision · Program_Chairs · 2023-09-21

**Decision:**

Accept (poster)

**Comment:**

This paper proposes a meta-learning method for hyperbolic graph neural networks. The proposed method uses hyperbolic meta gradients and label hyperbolic prototypes for effective meta-learning. The proposed approach is a novel combination of hyperbolic geometry and meta-learning techniques. The proposed method is scalable. The theoretical justification of the proposed method is also provided. The extensive experiments demonstrate the effectiveness of the proposed method. It would be good to add more explanation to clarify the contribution of the proposed method compared with existing work. Please improve the paper further according to the reviewers’ comments as responded in the author response.